# A Novel Rotational Field Eddy Current Planar Probe with Two-Circular Sector Pickup Coils

**DOI:** 10.3390/s19214628

**Published:** 2019-10-24

**Authors:** Guolong Chen, Weimin Zhang, Wuyin Jin, Weihan Pang, Zheng Cao, Kang Wang, Zhibo Song

**Affiliations:** 1School of mechanical and electrical engineering, Lanzhou university of technology, Lanzhou 730050, China; wuyinjin@hotmail.com (W.J.); dz82390q@163.com (Z.C.); 18686942382@139.com (K.W.); songzhibo0724@163.com (Z.S.); 2School of mechanical Engineering, Beijing institute of technology, Beijing 100081, China; 3Beijing Key Laboratory of Civil Aircraft Structures and Composite Materials, Beijing Aeronautical Science & Technology Research Institute of COMAC, Future Science and Technology Park, Changping, Beijing 102211, China; pangweihan@comac.cc

**Keywords:** nondestructive testing, rotational field, planar eddy current probe, eddy current testing

## Abstract

A new rotational field planar eddy current probe is proposed. The probe is combined with two orthogonal driver traces and a pickup coil that includes two-circular sector windings with series connection. Rotational eddy currents are induced by driver traces of the same amplitude and frequency, but fed with 90° phase different alternating exciting currents. An experimental demonstration using prototypes of the probes and artificial defects showed that the probe with a two-circular sector pickup coil is more sensitive for detecting the short defects than the probe with a circular pickup coil.

## 1. Introduction

Recently, the planar eddy current probe has become a research hotspot because it can be shaped to adapt metallic part surfaces to suppress lift-off noise and detect hard-to-reach areas [1,2]. Many structures have been proposed to design the planar flexible eddy current probe, such as the meandering winding magnetometer [3,4,5], rosette-like [6,7], parallelogram [8], mesh-winding [9], line [10], double rectangle [11], uneven step distribution [12], double-S shape [13], circular spiral [14], same direction line [4] and so on. Those structures are composed of line, circle and their spiral, so the eddy current in metallic parts with smaller direction range cannot be disturbed by the short defects in the same certain direction. 

To solve this problem, researchers have proposed many new structures for adjusting the eddy current distribution. Weimin Zhang proposed the fractal Koch curve structure to be an exciting coil-inducing eddy current distribution in many directions in a small region [15]. Then, in order to quantitatively evaluate the eddy current distribution, the angular spectrum and other indices based on information entropy have been proposed [16,17,18]. Those methods can be thought of as a space domain method. Still others are the time domain methods.

In the time domain methods, the idea of rotational field has been used to design an eddy current probe. The earliest rotational eddy current probe has been proposed by Hoshikawa, H. in 2006 [19]. Chaofeng Ye proposed a differential eddy current sensor with rotational magnetic field to evaluate multilayer structures [20]. Then, researchers proposed many different rotational field eddy current probes. However, until 2015, Rosado, Luis S. and et al. [21] proposed a dual driver traces planar eddy current probe, which is the first planar rotational field eddy current probe probe with driver and pickup coils. In the literature, the capacity of the planar rotational field eddy current probe was proved by simulation and experiment methods. In the same year, the planar rotational eddy current probe with two driver traces and a circular spiral pickup coil was proposed in our previous work. However, there remains a need for an efficient pickup coils that can effectively sense the eddy current disturbance by short defect in the different direction. 

The purpose of this study is to describe and examine two-circular sector pickup coils for two driver traces rotational eddy current probes. This paper is organized as follows: Section 1 gives a brief overview of the planar eddy current probes; Section 2 presents the probe principle and an assumption of two-circular sector pickup coils for the two driver traces rotational eddy current probes; in Section 3, the experimental setup is presented; in Section 4, the results are presented and analyzed; Conclusions are addressed in last section.

## 2. Probe Principle and Assumption

A mutual inductance planar eddy current probe includes an exciting element and a pickup element. In order to establish a primary magnetic field to induce the different eddy current distribution in conductive parts, the exciting element adopts the different types of winding, such as square and circular spiral and line segment. Then, coils are used to be a pickup element of the probe in this study. For the design of an eddy current probe, there are three questions to consider: (1)How does an eddy current probe generate an eddy current distribution which can be easily disturbed by defect?(2)How can an eddy current probe suppress the lift-off noise?(3)How can an eddy current probe sense the defect signal effectively?

Question 1 is about exciting coils of the eddy current probe. In this paper, the exciting traces include two line segments, which are perpendicular to each other. If the two alternating currents, which have the same amplitude and frequency but 90° phase difference, are fed into two drive traces, a rotational eddy current, which can be disturbed by the defect in any direction, will be induced in the conductive parts.

For Question 2, there are two methods to suppress the lift-off noise: signal processing [22] and special structure of the coils of the probe. In the literature [23], the authors proposed a method using the reference signals of the eddy current sensors in the air and on a perfect sample to reduce the lift-off effects. This algorithm is a signal processing approach. When an absolute eddy current sensor detects a short defect, the defect signal may be much weaker than the signal output from the eddy current sensor in the air or on a perfect testing part. Thus, in this case, the percentage of defect signal in the output signal is very small, which limits the gain of the signal conditional circuit. However, the reasonable coils’ structure design, such as differential structure, can lift the percentage [6,24]. The differential coils’ structure can cancel part of signal output from eddy current sensor in the air or on perfect testing samples. Thus, the differential pickup coils are used in this paper.

For Question 3, the pickup coils of the planar rotational field eddy current probe employ circular spiral coils, which were proposed and patented by our previous work, and use four-quadrant fan spiral coils proposed by [21]. In this work, two-circular sector pickup coils placed in the vertical angles of the driver traces are proposed, as shown in Figure 1a.

There is an assumption that the proposed pickup coil can pick up defect signals more effectively than the one winding of circular coil, whose radius is same as the radius of the circular sector of the proposed pickup coil. The reason of the assumption is shown in Figure 2. When there is not a defect in the specimen, the undisturbed line eddy currents in the two kinds of pickup coils will result in no output signal, because the magnetic flux density in opposite direction in the pickup coil are canceled by each other, as shown in Figure 2a,c. When a short defect in the specimen disturbs the eddy currents as shown in Figure 2b,d, the effect of the cancelling of the magnetic flux density in the two-sector pickup coils is weaker than that in the circular pickup coils, and the output signal of the two-sector pickup coil is stronger than that of circular pickup coils. Thus, for the two-driver traces eddy current probes, the proposed pickup coils with a half-area of the circular coil may be more sensitive than circular coils.

## 3. Experimental Setup

### 3.1. Eddy Current Probe

In order to do an experimental comparison study, two probes are designed and manufactured. The two probes have same exciting traces but different pickup coils, as shown in Figure 3. The exciting traces are two short line segments that are perpendicular to each other, and placed in the second and third layers of the print circuit board, respectively. The width of each exciting traces is 0.31 mm. The shape of pickup coils of the two-probe are two circular sectors and a circle, respectively. There is one turn in each pickup coil, and the width of the pickup coils’ winding is 0.15 mm. The pickup coils of each probe are placed in the fourth layer of the print circuit board. 

The probe with two circular sectors pickup coils is called probe A, but the other one is called probe B.

### 3.2. Specimen

The artificial defects are slots manufactured by electrical discharge machining in aluminium plates 6 mm in depth. The first defect is a slot whose size is 0.25 mm × 80 mm × 3 mm on a 350 mm × 80 mm × 6 mm size aluminium specimen. The length of this defect equals the length of the specimen, and the defect—which is named infinite defect—cuts off the specimen. For surveying the response of different direction defect, 5.0 mm × 0.1 mm × 1.0 mm size defect is produced on the specimen with the rotation of the electric control rotary table. 

To evaluate the resolution ability of probe B, two sets of slot defects are prepared. The first set of defects have a different length but same depth and width, whose lengths are 1.0 mm, 3.0 mm, 5.0 mm, 10 mm, 15 mm, 20 mm, respectively, and whose width and depth are 0.22 mm and 1.0 mm, respectively. The second set of defects have a different depth, which varies from 0.1 mm to 1 mm with the step of 0.1 mm. For each depth, the defect has lengths of 5 mm and 10 mm and the same width of 0.22 mm.

### 3.3. Experimental System

For the experiment to evaluate the proposed probe and assumptions about it, an experimental system is set up, as shown in Figure 4. The experiment includes a signal generator (GD1022, RIGOL Technologies, Inc., Jiangsu, China), two power amplifiers (LPA05A, Newtons4th Ltd., Leicester, UK), a multi-meter (DM3058, RIGOL Technologies, Inc.), probes, signal conditioning circuit, an oscilloscope (DS1102E, RIGOL Technologies, Inc.), a computer, a three-dimensional scanning station, and an electric control rotary table. 

Two alternating currents with 0.2 A Root mean square (RMS) and 100 kHz frequency but 90° phase difference are feed to the two driver traces, respectively, to generate alternating magnetic field and then the eddy currents are induced in the specimen to detect the defect. The conditioning circuit, which is an I/Q demodulation, extracts the real and imaginary part of the signal output from the pickup coil. Then, the signals are observed and saved by oscilloscope. The saved signals are reprocessed by a 100-Hz low-pass filter in MATLAB.

The motion of the probes is controlled by the three-dimensional scanning station by the computer, and the rotational motion of the specimen with the electric control rotary table is controlled manually.

The experiments include two sets. The first set are contrast experiments of the two probes. In this experiment, the 5 mm × 0.1 mm × 1 mm defect in different directions and the infinite length defect are detected. The second set of experiments are capability evaluation experiments for probe A. The aforementioned different length and depth defects are detected.

For all experiments, the probes move in the direction of internal bisector of two driver traces in which the pickup coils of probe A are placed.

## 4. Result and Discussion

### 4.1. Contrast Experiment of the Two Probes

Figure 5 shows the signal of the infinite defect output from two probes. The waveform of real and imaginary parts of probe A are in antiphase, but those of probe B are in phase. The peak-to-peak value (*V_pp_*) of the real part signals of probe A and B are 0.71 V and 0.35 V, respectively, and the *V_pp_* of the imaginary part of probe A and B are 0.51 V and 0.52 V, respectively. Thus, the $V_{pp}$ of the real part of probe A is about twice as large as that of probe B, and the *V_pp_* of the imaginary part of probe A is about 1.22 times larger than that of probe B.

Figure 6 and Figure 7 show the signals’ output from the two probes scanning the 5-mm length defect in different directions. For probe A, the variation trends of real and imaginary parts are the same. Then, the real part and imaginary parts are in antiphase for a certain defect direction. However, for probe B, the variation trend of 0°, 15°, 30° directions of defect are the same but opposite to the rest of the directions of the defect.

Figure 8a shows the $V_{pp}$ of the real and imaginary parts of the two probes for different direction of the 5-mm length defect. The signals of probe A are larger than that of probe B. Figure 8b shows the rate of change of probe A are relative to B for a certain direction of defect. The rates of change of real and imaginary parts in all directions are larger than 0. For the real part of the rate of change, the maximum value is 829% at 60° defect direction, but the minimum is 232% at 15° defect direction. For the imaginary parts of the rate of change, the maximum value is 668% at 60° defect direction, but the minimum is 184% at 90° defect direction.

To sum up, for the 5-mm length defect in any direction, the sensitivity of probe A is significantly higher than that of probe B, although the area of the pickup coil of probe A has half that of probe B. The reason for this phenomenon is that the circular pickup coil cannot effectively sense the eddy current defect signal because the magnetic field in opposite direction of disturbed eddy currents are mainly cancelled by each other. Thus, the two-sector pickup coils can more effectively extract the disturbed eddy currents induced by two orthogonal driver traces than the circular pickup coil of probe B.

### 4.2. Capability Evaluation Experiment

Table 1 and Table 2 show the *V_pp_* of the real and imaginary parts signal for different depth defect. The two part signals of 5-mm and 10-mm length defects increase approximately as the depth of defect increase. For the each certain defect, the real part of *V_pp_* is larger than the imaginary part. The maximum values of the real and imaginary parts of the 5-mm length defects are 0.376 V and 0.159 V, respectively. Those values of the 10-mm length defects are 0.498 V and 0.294 V, respectively, which are larger than those of the 5-mm length defects.

Figure 9 shows the two parts of signal of different length defects. On the whole, the real part of the signal for the certain defect is larger than the imaginary part. When the length of the defect is smaller than 10 mm, the two-part signals increase as the defect length increases; however, when the length of the defect is longer than 10 mm, the two-part signals have little change with the change of the defect length. This phenomenon means that when the defect is larger than the size of pickup coils over the center of the defect, the disturbed eddy currents under the pickup coil do not change with the change of the defect length. In other words, the pickup coil “sees” the same magnetic field distribution for this situation. Thus, the output signals do not change when the defect length is larger than 10 mm, which equals the diameter of the pickup coil.

## 5. Conclusions

In this paper, the two-circular sector pickup coil is proposed to effectively boost the sensitivity of the two driver traces rotational field planar eddy current probe. The assumption is made that the proposed pickup coils are more sensitive than the same diameter circular pickup coils of the two driver traces probe detecting the short defect. The contrast experiment and the capability evaluation experiment are conducted. The experimental results show that probe A with the proposed pickup coil is more sensitive than probe B with the circular pickup coil. In any defect direction, the output signal of probe A is especially larger than that of probe B. Thus, the hypothesis is verified by the experiment. 

In future work, another two-sector circular pickup coil will be added in the rest of the vertical angles of the two-driver trace, and the defect signals output from it will be studied. Moreover, the quantitative recognition algorithm of the proposed probe will be studied as well.

## Figures and Tables

**Figure 1 sensors-19-04628-f001:**
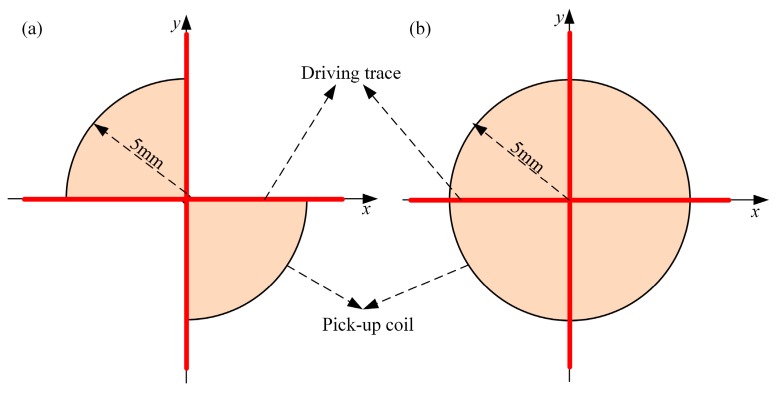
Two pickup coils for the two driver traces perpendicular to each other: (**a**) circular pickup coil, (**b**) two-circular sector pickup coil.

**Figure 2 sensors-19-04628-f002:**
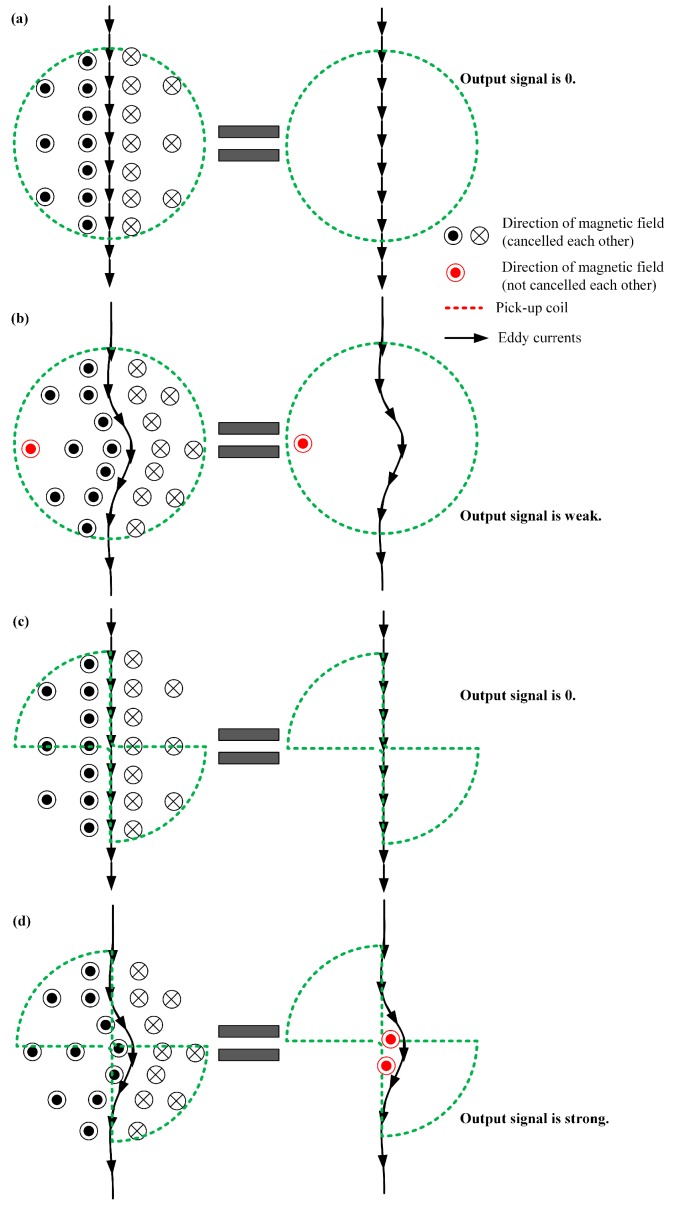
Eddy current disturbance: (**a**) no disturbance in the circular pickup coil, (**b**) disturbance in the circular pickup coil, (**c**) no disturbance in the two-circular sectors pickup coil, (**d**) disturbance in the two-circular sectors pickup coil.

**Figure 3 sensors-19-04628-f003:**
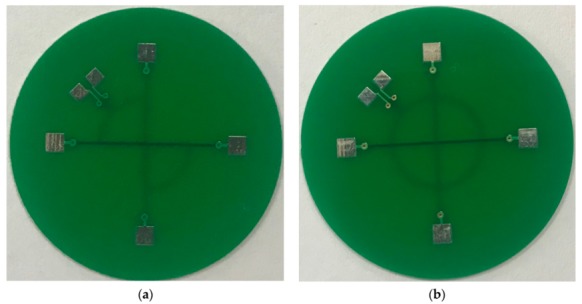
The two kinds of probes: (**a**) probe A, (**b**) probe B.

**Figure 4 sensors-19-04628-f004:**
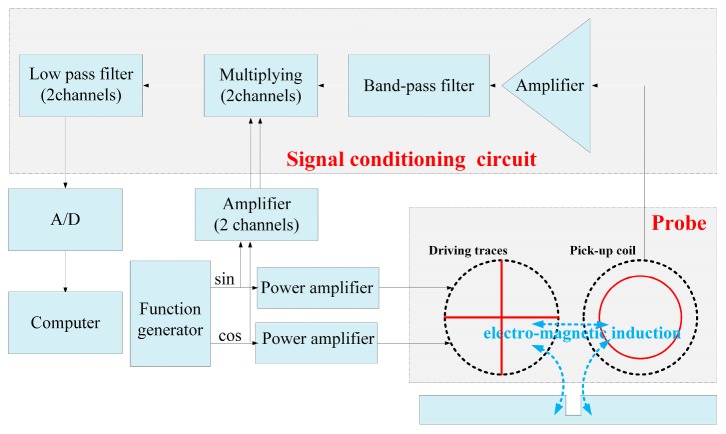
The schematic diagram for experimental system.

**Figure 5 sensors-19-04628-f005:**
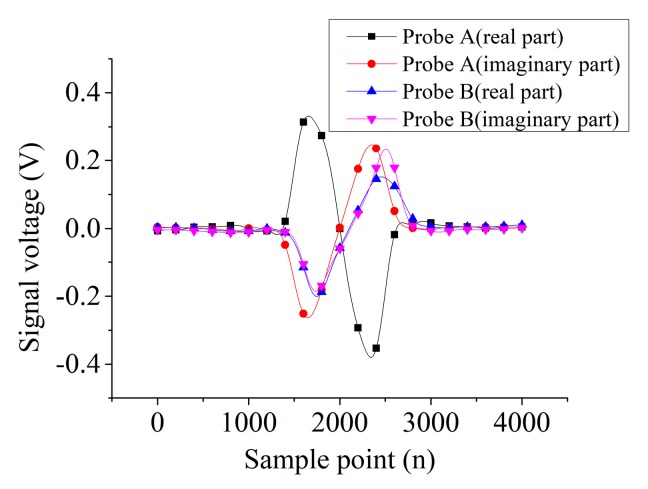
The real and imaginary part signal for infinite defect.

**Figure 6 sensors-19-04628-f006:**
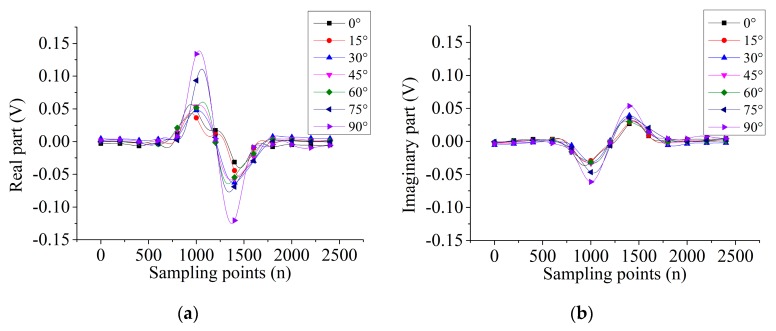
The signal output of 5-mm length defect by probe A: (**a**) real part, (**b**) imaginary part.

**Figure 7 sensors-19-04628-f007:**
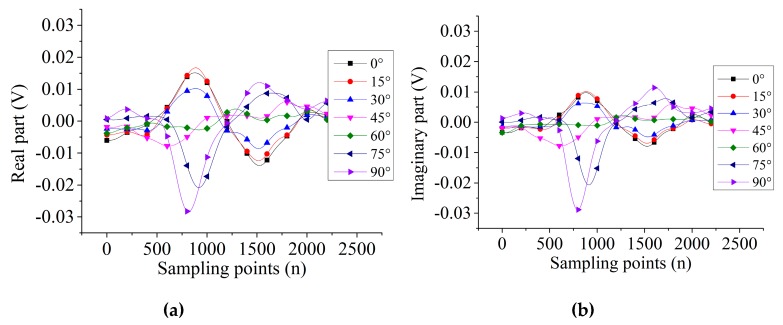
The signal output of 5-mm length defect by probe B: (**a**) real part, (**b**) imaginary part.

**Figure 8 sensors-19-04628-f008:**
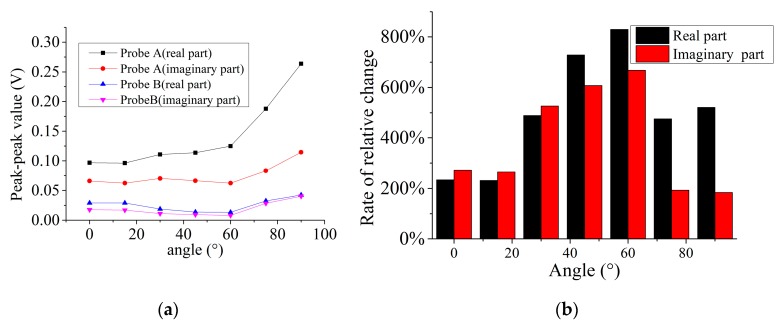
Peak contrast of the signals of two probes: (**a**) peak-to-peak value, (**b**) rate of change of peak-to-peak value.

**Figure 9 sensors-19-04628-f009:**
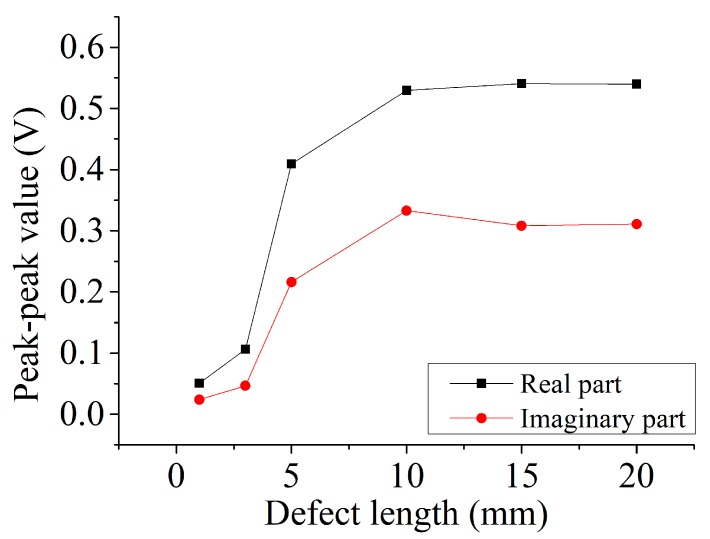
Real and imaginary part output from probe A of different length defects.

**Table 1 sensors-19-04628-t001:** Real and imaginary part output from probe A of 5-mm length but different depth defects.

Defect Depth (mm)	Real Part *V_pp_* (V)	Imaginary Part *V_pp_* (V)
0.1	0.077	0.016
0.2	0.133	0.036
0.3	0.229	0.076
0.4	0.218	0.081
0.5	0.255	0.103
0.6	0.295	0.125
0.7	0.272	0.119
0.8	0.355	0.152
0.9	0.369	0.160
1.0	0.376	0.159

**Table 2 sensors-19-04628-t002:** Real and imaginary part output from probe A of 1-mm length but different depth defects.

Defect Depth (mm)	Real Part *V_pp_* (V)	Imaginary Part *V_pp_* (V)
0.1	0.044	0.014
0.2	0.131	0.051
0.3	0.187	0.084
0.4	0.216	0.106
0.5	0.319	0.165
0.6	0.378	0.201
0.7	0.396	0.219
0.8	0.488	0.266
0.9	0.473	0.281
1.0	0.498	0.294

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
