# Peer review of "A Novel Rotational Field Eddy Current Planar Probe with Two-Circular Sector Pickup Coils"

_sensors, 2019, doi:10.3390/s19214628_

Round 1

Reviewer 1 Report

The work is interesting. However, the presentation is poor. Significant improvement is required as suggested below.

The research motivation and background with reference are required. e.g. 

Chaofeng Ye etc.,Differential Sensor Measurement With Rotating Current Excitation for Evaluating Multilayer Structures,IEEE Sensors Journal 16(3):1-1 · January 2015. DOI: 10.1109/JSEN.2015.2488289

Damhuji Rifai etc., Giant Magnetoresistance Sensors: A Review on Structures and Non-Destructive Eddy Current Testing Applications, Sensors 2016, 16(3), 298; https://doi.org/10.3390/s16030298

DI Ona, GY Tian, R Sutthaweekul, SM Naqvi, Design and optimisation of mutual inductance based pulsed eddy current probe, Measurement 144, 402-409, 2019.

 In section 1 line 1, the authors stated ‘’A mutual inductance planar eddy current probe concludes an exciting element and a pick-up element’’. Do the authors mean conclude or include? Also a capital letter should be used after a full stop. Write’ In order’ not ‘in order’ after the full stop. In section 1 paragraph 2, the authors used alternative current. Do the authors want to write alternating current? More explanation is required. In section 1 paragraph 3, the authors stated that both signal processing and special structure of the coils of eddy current probe can be used to reduce lift-off effects (noise0. However the authors used differential pick-up coil to suppress lift-off effects without explaining why coil structure is used instead of signal processing approach. The authors should explain why coil structured is used instead of signal processing technique including normalisation and separation with reference.  In section 1 paragraph 4, the authors explained the principle of the circular pick-up coil and why it is not sensitive but little is said about the proposed circular-sector pick-up coil and why its sensitivity is higher. More explanation of the proposed pick-up and how the improvement in its sensitivity come about is needed.  The authors based the experimental results and comparisons on amplitude of the response signals. The reference subtracted signal should be used and not mere amplitude value. The reference being the probe in air, without the effect of the conductor.  The last sentence of section 4.1 the authors stated ‘Thus, the two sectors pick-up coils can more effectively abstract the disturbed eddy current induced by two orthogonal driver traces than the circular pick- up coil of the probe B’. The word ‘abstract’ as used in the sentence is not clear. It should be explained. Figures 1-3 should have better plotted. Configuration of coils with pictures should be provided. More critical discussion and logic links could be improved. English proof read is also expected.

Author Response

The work is interesting. However, the presentation is poor. Significant improvement is required as suggested below.

The research motivation and background with reference are required. e.g. 

Chaofeng Ye etc.,Differential Sensor Measurement With Rotating Current Excitation for Evaluating Multilayer Structures,IEEE Sensors Journal 16(3):1-1 · January 2015. DOI: 10.1109/JSEN.2015.2488289

Damhuji Rifai etc., Giant Magnetoresistance Sensors: A Review on Structures and Non-Destructive Eddy Current Testing Applications, Sensors 2016, 16(3), 298; https://doi.org/10.3390/s16030298

DI Ona, GY Tian, R Sutthaweekul, SM Naqvi, Design and optimisation of mutual inductance based pulsed eddy current probe, Measurement 144, 402-409, 2019.

Responding: Thank you for reviewer’s suggestion. We added the three references in the manuscript.

 In section 1 line 1, the authors stated ‘’A mutual inductance planar eddy current probe concludes an exciting element and a pick-up element’’. Do the authors mean conclude or include? Also a capital letter should be used after a full stop. Write’ In order’ not ‘in order’ after the full stop. In section 1 paragraph 2, the authors used alternative current. Do the authors want to write alternating current?

Responding: Thank you for reviewer’s suggestion. I am sorry for these mistake. We corrected the mistake in the manuscript.

More explanation is required. In section 1 paragraph 3, the authors stated that both signal processing and special structure of the coils of eddy current probe can be used to reduce lift-off effects noise. However the authors used differential pick-up coil to suppress lift-off effects without explaining why coil structure is used instead of signal processing approach. The authors should explain why coil structured is used instead of signal processing technique including normalisation and separation with reference.

Responding: Thanks for reviewer’s suggestion about lift-off noise. The reasons of using coils structure instead of signal processing approach as described in section 1 paragraph 3: “In literature the authors prosed a method using the reference signals of eddy current sensors in the air and on a perfect samples to reduce the lift-off effects. This method is a signal processing approach. When an absolute eddy current sensor detects a short defect, defect signal may much weaker than signals output from eddy current sensor in the air or on a perfect testing part. Thus, in this case, the percentage of defect signal in the output signal is very small, which limits the gain of signal conditional circuit. However, reasonable coils structure design, such as differential structure, can lift the percentage. The differential coils structure can cancel part of signal output from eddy current sensor in the air or on the perfect testing samples. Thus, in this paper, the differential pick-up coils are used.”

 In section 1 paragraph 4, the authors explained the principle of the circular pick-up coil and why it is not sensitive but little is said about the proposed circular-sector pick-up coil and why its sensitivity is higher. More explanation of the proposed pick-up and how the improvement in its sensitivity come about is needed.

Responding: Thanks for the reviewer’s suggestion. The reason of the proposed coils structure improve sensor’s sensitivity is as follow:

The eddy currents induced by the exciting coils distributes in the round of the circular pick-up coils projected to the sample. When the defect is shorter than the diameter of circular coil, the disturbed eddy currents are in the pick-up coils and the opposite magnetic field induced by eddy disturbed eddy current are cancel each other, so the circular coil can not effectively extract the defect signal. The proposed two circular sector pick-up coils can separate the two opposite direction magnetic fields as much as possible. This reason is described in section 1 paragraph 5.

 The authors based the experimental results and comparisons on amplitude of the response signals. The reference subtracted signal should be used and not mere amplitude value. The reference being the probe in air, without the effect of the conductor.  

Responding: Thanks for the reviewer’s suggestion. Because the probe used the differential structure, so, under the ideal conditions, when the probe in the air and on a perfect sample, the output signal is 0. Thus, the  for traditional eddy current sensor dose not suite to the differential sensors. To test the hypothesis of this article, we use same exciting alternating current, same signal conditional circuit and same defects, which are fair to the probe in this paper.

The last sentence of section 4.1 the authors stated ‘Thus, the two sectors pick-up coils can more effectively abstract the disturbed eddy current induced by two orthogonal driver traces than the circular pick- up coil of the probe B’. The word ‘abstract’ as used in the sentence is not clear. It should be explained. Figures 1-3 should have better plotted. Configuration of coils with pictures should be provided. More critical discussion and logic links could be improved. English proof read is also expected. 

Responding: Thanks for the reviewer’s suggestion. I am sorry for the mistakes and we corrected them. The word “abstract” is corrected to “extract”. Figures 1-3 are plotted again. Then, we improved the discussion and logic of this paper. We carefully corrected the English language of this manuscript.

Reviewer 2 Report

Dear Author,

the paper covers in principle an interesting study that's beneficial to the Eddy Current community.

By normal reading the accessibility to the papers content is difficult due to   very lean illustration's in Fig 1 and Fig.3 and to compact description of the electrical circuit of the coil configuration.

The reviewer strongly recommend to improve this drawings with axle description and more detailed  information.  Please extent the document by a photograph or sketch of the real probe used for the experiment.

Author Response

the paper covers in principle an interesting study that's beneficial to the Eddy Current community. By normal reading the accessibility to the papers content is difficult due to very lean illustration's in Fig 1 and Fig.3 and to compact description of the electrical circuit of the coil configuration. The reviewer strongly recommend to improve this drawings with axle description and more detailed information.  Please extent the document by a photograph or sketch of the real probe used for the experiment.

Responding: Thanks for reviewer’s suggestion. We redraw Fig.1 and Fig.3, and added the picture of real probes.

Reviewer 3 Report

Authors in this paper are presenting a realization of an eddy current probe of the kind of rotating magnetic field consisting of a common configuration of a pair of crossing conductors in quadrature current excitation and a pair of coplanar circular sectors to pick-up the signal. In section 2, they give intuitive arguments about superior sensitivity of their proposed two circular sectors pick-up configuration as compared to a single spiral circular coil that are trying to show experimentally in the next sections of their paper.

Reviewer’s opinion is that the proposed probe configuration neither constitute any novelty in the area, as many other interesting variations of differential probes have been appeared years now in the literature, nor its design was thoroughly studied by the authors comparatively to others by means of any kind of simulation or even experimentally. Besides, authors’ comments on the results of their limited experimental repertoire consist of commonplaces with no attempt to attribute the underlying physical meanings just confined to describe figures. Finally, English language writing is a serious drawback and should be reconsidered by authors team in any of their future submission.

I regret to suggest this work not to be considered for publication in this form.

Author Response

Authors in this paper are presenting a realization of an eddy current probe of the kind of rotating magnetic field consisting of a common configuration of a pair of crossing conductors in quadrature current excitation and a pair of coplanar circular sectors to pick-up the signal. In section 2, they give intuitive arguments about superior sensitivity of their proposed two circular sectors pick-up configuration as compared to a single spiral circular coil that are trying to show experimentally in the next sections of their paper.

Reviewer’s opinion is that the proposed probe configuration neither constitute any novelty in the area, as many other interesting variations of differential probes have been appeared years now in the literature, nor its design was thoroughly studied by the authors comparatively to others by means of any kind of simulation or even experimentally. Besides, authors’ comments on the results of their limited experimental repertoire consist of commonplaces with no attempt to attribute the underlying physical meanings just confined to describe figures. Finally, English language writing is a serious drawback and should be reconsidered by authors team in any of their future submission.

I regret to suggest this work not to be considered for publication in this form.

Responding: Thanks for reviewer’s comments for this manuscript. For the planar eddy current probe whose exciting and sensing element adopt coils, there is one literature reporting the rotational field eddy current probe.

“Rosado, L. S.; Santos, T. G.; Ramos, P. M.; Vilaça, P.; Piedade, M., A new dual driver planar eddy current probe with dynamically controlled induction pattern. NDT and E International 2015, 70, 29-37.” 

In this paper, the authors demonstrated that when the two exciting currents with same amplitude but  phase difference feed to two perpendicular exciting driving trace, the rotational eddy current probe can effectively detect the defects in different direction. However, the defect length 20mm is larger than total size of pick-up coils 10mm. If we used a traditional circular eddy current probe, when the defect length is larger than the diameter of the probe coils, eddy current can be disturbed by any direction defect. The advantage of rotational eddy current probe is to detect the short defect. Our work uses the defects whose length is short than that of diameter of pick-up coil. Then, in the paper “Rosado, L. S.; Santos, T. G.; Ramos, P. M.; Vilaça, P.; Piedade, M., A new dual driver planar eddy current probe with dynamically controlled induction pattern. NDT and E International 2015, 70, 29-37.” the four pick-up coils are in series and form a differential structure to suppress the lift-off noise. However, when the short defect disturbed eddy current, are the signal of the defect output from four coils constructive or destructive? Our work separates four coils and mere use two coils, and we compared the proposed pick-up coils and one circular pick-up coils.

Reviewer 4 Report

A mutual inductance planar eddy current probe concludes an exciting element and a pick-up 57 element. in order - missing capital letter

When no defect is in the 82 specimen, an un-disturbed line – use another word

When no defect is in the 82 specimen, an un-disturbed line eddy current in the circular pick-up coil results in no output signal Sensors 2019, 19, x FOR PEER REVIEW 3 of 9 83 because the magnetic flux density in opposite direction in the pick-up coil are canceled each other, 84 shown in figure 2(a); when a short defect in the specimen disturb the eddy current as shown in figure 85 2(b), the symmetry of secondary magnetic is broken, but the strongest magnetic flux density in 86 opposite direction is still in the circular and canceled each other. – very complicated formulation without clear meaning

The width of each exciting traces 98 0.305mm(12mil). – what does it mean: 12mil?

0.152mm  - too precise. it is sufficient to use two decimal points

The artificial defects are slots – better to use notches instead of slots

3.3 experiment system – capital letter

The experiment includes a signal generator, two power amplifier, a multi-meter, 123 probes, signal conditioning circuit, an oscilloscope, a computer, a three-dimension scanning station 124 and an electric control rotary table. – please state concrete models  and use plural

Two alternative current with 0.2 A RMS and 100 KHz frequency but 90° phase difference are 126 feed to the two driver traces, respectively, to generate alternative magnetic field and then the eddy 127 current is induced in the specimen to detect the defect. – eddy currents, plural

differential pick-up coil outputs the signal. – bad syntax and also the meaning

Then the conditioning circuit which is a I/Q demodulate extracts the real and imaginary part of the 129 signal. – incorrect verb used

For all experiment, the probes move in the direction of internal bisector of two driver traces in 141 which the pick-up coil of probe A are placed. – please correct to plural

4.1 Contrast experiment – ambiguous title

The peak-peak value 146 (???) of the real part signals of probe A and B are 0.71 V and 0.35 V, respectively, and the ??? of the 147 imaginary part of probe A and B are 0.51 V and 0.52 V, respectively. T – please correct peak-to-peak

Figure 7. – horizontal axis: please change sampling to distance

Fig 8 – rate of relative change: the unit is VOLTS, there is a mistake! what does it mean 800% at 60 degrees angle?

Figure 9. R – individual points can not be linked by the line. please use regression

Thus, the output signals almost don’t change when the defect 201 length is larger than 10mm which equals to the dimeter of the pick-up coil. – keying mistake in the word of diameter

Author Response

A mutual inductance planar eddy current probe concludes an exciting element and a pick-up 57 element. in order - missing capital letter

Responding: Thanks for reviewer’s suggestion. I am sorry to out mistake, we corrected it.

When no defect is in the  specimen, an un-disturbed line – use another word

Responding: we used the “straight line” instead of “un-disturbed line”.

When no defect is in the specimen, an un-disturbed line eddy current in the circular pick-up coil results in no output signal because the magnetic flux density in opposite direction in the pick-up coil are canceled each other, shown in figure 2(a); when a short defect in the specimen disturb the eddy current as shown in figure 2(b), the symmetry of secondary magnetic is broken, but the strongest magnetic flux density in opposite direction is still in the circular and canceled each other. – very complicated formulation without clear meaning

Responding: Thanks for reviewer’s suggestion. we revised this paragraph to “When there is not a defect in the specimen, un-disturbed line eddy currents in the circular pick-up coil will results in no output signal because the magnetic flux density in opposite direction in the pick-up coil are canceled each other, shown in figure 2(a); when a short defect in the specimen disturb the eddy currents as shown in figure 2(b), much of the magnetic flux density induced by eddy currents canceled each other, and the probe can output a very weak defect signal.”

The width of each exciting traces 0.305mm(12mil). – what does it mean: 12mil?

0.152mm - too precise. it is sufficient to use two decimal points

Responding: Thanks for reviewer’s suggestion. We correct the 0.305mm to 0.31mm, and the 0.152mm to 0.15mm. Then we deleted the 12 mil which is a imperial unit.

The artificial defects are slots – better to use notches instead of slots

3.3 experiment system – capital letter

Responding: Thanks for reviewer’s suggestion. We corrected this mistake.

The experiment includes a signal generator, two power amplifier, a multi-meter, probes, signal conditioning circuit, an oscilloscope, a computer, a three-dimension scanning station and an electric control rotary table. – please state concrete models and use plural

Responding: Thanks for reviewer’s suggestion. We the concrete models in the manuscript. “The experiment includes a signal generator (GD1022, RIGOL Technologies, Inc.), two power amplifiers (LPA05A, Newtons4th Ltd.), a multi-meter (DM3058, RIGOL Technologies, Inc.), probes, signal conditioning circuit, an oscilloscope (DS1102E, RIGOL Technologies, Inc.), a computer, a three-dimension scanning station and an electric control rotary table.”

Two alternative current with 0.2 A RMS and 100 KHz frequency but 90° phase difference are  feed to the two driver traces, respectively, to generate alternative magnetic field and then the eddy 127 current is induced in the specimen to detect the defect. – eddy currents, plural

Responding: thanks for reviewer’s suggestion. We corrected it.

differential pick-up coil outputs the signal. – bad syntax and also the meaning

Then the conditioning circuit which is a I/Q demodulate extracts the real and imaginary part of the signal. – incorrect verb used

Responding: Thanks for reviewer’s suggestion. We correct this two sentence to “The conditioning circuit which is a I/Q demodulation extracts the real and imaginary part of the signal output from the pick-up coil.”

For all experiment, the probes move in the direction of internal bisector of two driver traces in which the pick-up coil of probe A are placed. – please correct to plural

Responding: Thanks for reviewer’s suggestion. We correct it.

4.1 Contrast experiment – ambiguous title

Responding: Thanks for reviewer’s suggestion. We correct “Contrast experiment” to “Contrast experiment of the two probes”

The peak-peak value 146 (???) of the real part signals of probe A and B are 0.71 V and 0.35 V, respectively, and the ??? of the 147 imaginary part of probe A and B are 0.51 V and 0.52 V, respectively. T – please correct peak-to-peak

Responding: Thanks for reviewer’s suggestion and we sorry to this mistake. We correct the “peak-peak” to “peak-to-peak”.

Figure 7. – horizontal axis: please change sampling to distance

Responding: Thanks for reviewer’s suggestion. In this study, we just compare the two kinds of probes, I think “sampling points ” will not affect the result.

Fig 8 – rate of relative change: the unit is VOLTS, there is a mistake! what does it mean 800% at 60 degrees angle?

Responding: we correct the unit of figure 8. The 800 at 60 degrees angle mean that when the defect direction is 60 degrees, the Vpp of probe A is larger than that of probe B. Thus, we correct all direction to defect direction.

Figure 9. R – individual points can not be linked by the line. please use regression

Responding: Thanks for reviewer’s suggestion. We use two tables instead of figure 9 and figure 10.

Thus, the output signals almost don’t change when the defect length is larger than 10mm which equals to the dimeter of the pick-up coil. – keying mistake in the word of diameter

Responding: I am sorry for our mistake. We correct it.

Round 2

Reviewer 1 Report

The improvement is not good enough. Further improvement is required.

Mind English proof read and format and consistent font; e.g.

Figure 1 and 2 should read as Figures 1 and 2;

Mind valid digits e.g. 1 should read as 1.0 in the Tables 1 and ; 5mm×0.1mm×1mm should read as 5.0mm×0.1mm×1.0mm etc;  Ref. 22 and 24 are incomplete to me; 4. Figure 2 and its explanation are trivial to me; Conclusion and further work need to be discussed and justified.

Author Response

Reviewer 1

Mind valid digits e.g. 1 should read as 1.0 in the Tables 1 and ; 5mm×0.1mm×1mm should read as 5.0mm×0.1mm×1.0mm etc; 

Reponse: thanks for reviewer’s suggestion. We correct the numbers and units carefully.

Ref. 22 and 24 are incomplete to me;

Reponse: thanks for reviewer’s suggestion. We correct the Ref. 22 and 24.

Ona, D. I.; Tian, G. Y.; Sutthaweekul, R.; Naqvi, S. M., Design and optimisation of mutual inductance based pulsed eddy current probe. Measurement 2019, 144, 402-409.

Santos, T. S.; Ramos, P. M.; Vilaça, P. S. In Non destructive testing of friction stir welding: Comparison of planar eddy current probes, Imeko Tc4 Symposium, 2008.

Figure 2 and its explanation are trivial to me;

Reponse: thanks for reviewer’s suggestion. We redesigned the figure 2 and explain the figure 2. In the new figure 2, we add another two parts of proposed probe to illustrate the eddy current disturbance in the two sectors circular pick-up coils. Then, the explanation of figure is changed to “When there is not a defect in the specimen, un-disturbed line eddy currents in the two kinds of pick-up coils will result in no output signal, because the magnetic flux density in opposite direction in the pick-up coil are canceled each other, shown in figure 2(a) and figure 2(c); when a short defect in the specimen disturb the eddy currents as shown in figure 2(b) and figure 2(d), the effect of the cancelling of the magnetic in the two sector pick- up coils is weaker than that in the circular pick-up coils, and the output signal of two sector pick-up coil is stronger than that of circular pick-up coils. Thus, for the two driver traces eddy current probe, the proposed pick-up coils with half area of the circular coil may have higher sensitivity than circular coils.”

Conclusion and further work need to be discussed and justified.

Reponse: thanks for reviewer’s suggestion. The future work of the proposed probe is added at the end of the manuscript “In the future work, another two sectors circular pick-up coils will be added in the rest vertical angles of the two driver trace, and the defect signals output from it will be studied. Moreover, the quantitative recognition algorithm of the proposed probe will be studied too.”

Reviewer 3 Report

This newer manuscript is actually identical to the one initially submitted. Comments on that previous manuscript about its novelty and the intuitive character of the arguments used by the authors to describe probe's design instead of using a mathematically rigid method (analytical or simulation) still hold. If we only concentrate on issues like the effort made by the team to manufacture the probe and its unambiguously higher signal to noise ration as compared to a common circular pick-up coil we could accept it for publication after only substituting as a horizontal axis variable of Figures 5, 6, and 7 "Sampling points (n)" with the position or scanning length (x) in mm. 

Author Response

Reponse: Thanks for reviewer suggestion. As your suggestion, a mathematically rigid method is the best way. However, there two reasons of no analytical and simulation in this manuscript: 1. The analytical or simulation method tell us that which kind of pick-up coil is the candidate, but it can only select it in the candidates. 2. The signal of the planar eddy current for detecting tiny defect is too weak to calculated by simulation method. We proposed the pick-up coils based on the understanding of the mechanism of eddy current probe. We think the mutual eddy current probe may sense the disturbance information effectively. This idea is illustrated in the figure 2.

Then, why we use the “Sampling points(n)” instead of the position is because that the probe continually moved on the scanning table and the AD card continually acquired the data instead of step by step scanning. And, the two probes use the totally same procedure in the experiment. I admit that the data in the figures 5, 6, 7 can not be used to quantitative recognition of defect size and position, but those data do not affect the results of the comparisons in this manuscript.

Thanks for reviewer’s suggestion again. We plan to study the defect quantitative recognition. In the future work, the data will be acquired by point to point, and the position information will be represented.

Reviewer 4 Report

The authors have taken into the consideration all of my suggestions, remarks and comments. From my point of view the article can be accepted as submitted. 

Author Response

The authors have taken into the consideration all of my suggestions, remarks and comments. From my point of view the article can be accepted as submitted. 

Reponse: Thank you for approving our job.